# VOS: Learning What You Don't Know by Virtual Outlier Synthesis

**Xuefeng Du, Zhaoning Wang, Mu Cai, Yixuan Li**
Department of Computer Sciences
University of Wisconsin - Madison
{xfdu,mucai,sharonli}@cs.wisc.edu

## Abstract

Out-of-distribution (OOD) detection has received much attention lately due to its importance in the safe deployment of neural networks. One of the key challenges is that models lack supervision signals from unknown data, and as a result, can produce overconfident predictions on OOD data. Previous approaches rely on real outlier datasets for model regularization, which can be costly and sometimes infeasible to obtain in practice. In this paper, we present **VOS**, a novel framework for OOD detection by adaptively synthesizing virtual outliers that can meaningfully regularize the model's decision boundary during training. Specifically, VOS samples virtual outliers from the low-likelihood region of the class-conditional distribution estimated in the feature space. Alongside, we introduce a novel unknown-aware training objective, which contrastively shapes the uncertainty space between the ID data and synthesized outlier data. VOS achieves competitive performance on both object detection and image classification models, reducing the FPR95 by up to 9.36% compared to the previous best method on object detectors. Code is available at https://github.com/deeplearning-wisc/vos.

## 1 Introduction

Modern deep neural networks have achieved unprecedented success in known contexts for which they are trained, yet they often struggle to handle the unknowns. In particular, neural networks have been shown to produce high posterior probability for out-of-distribution (OOD) test inputs (Nguyen et al., 2015), which arise from unknown categories and should not be predicted by the model. Taking self-driving car as an example, an object detection model trained to recognize in-distribution objects (*e.g.,* cars, stop signs) can produce a high-confidence prediction for an unseen object of a moose; see Figure 1(a). Such a failure case raises concerns in model reliability, and worse, may lead to catastrophe when deployed in safety-critical applications.

The vulnerability to OOD inputs arises due to the lack explicit knowledge of unknowns during training time. In particular, neural networks are typically optimized only on the in-distribution (ID) data. The resulting decision boundary, despite being useful on ID tasks such as classification, can be ill-fated for OOD detection. We illustrate this in Figure 1. The ID data (gray) consists of three class-conditional Gaussians, on which a three-way softmax classifier is trained. The resulting classifier is overconfident for regions far away from the ID data (see the red shade in Figure 1(b)), causing trouble for OOD detection. Ideally, a model should learn a more compact decision boundary that produces low uncertainty for the ID data, with high OOD uncertainty elsewhere (*e.g.*, Figure 1(c)). However, achieving this goal is non-trivial due to the lack of supervision signal of unknowns. This motivates the question: *Can we synthesize virtual outliers for effective model regularization?*

In this paper, we propose a novel unknown-aware learning framework dubbed **VOS** (**V**irtual **O**utlier **S**ynthesis), which optimizes the dual objectives of both ID task and OOD detection performance. In a nutshell, VOS consists of three components tackling challenges of outlier synthesis and effective model regularization with synthesized outliers. To synthesize the outliers, we estimate the class-conditional distribution in the *feature space*, and sample outliers from the low-likelihood region of ID classes (Section 3.1). Key to our method, we show that sampling in the feature space is more tractable than synthesizing images in the high-dimensional pixel space (Lee et al., 2018a).

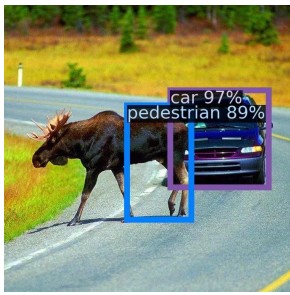
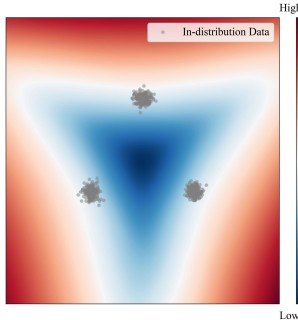
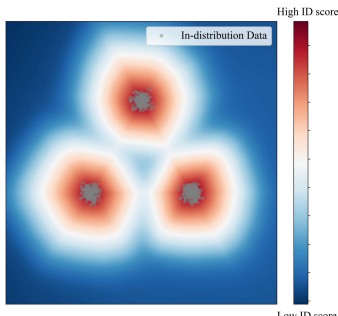

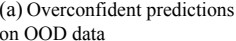
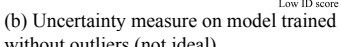

(a) Overconfident predictions on OOD data  (b) Uncertainty measure on model trained without outliers (not ideal)  (c) Uncertainty measure on model trained with virtual outliers (ours)

Figure 1: (a) A Faster-RCNN (Ren et al., 2015) model trained on BDD-100k dataset (Yu et al., 2020) produces overconfident predictions for OOD object (*e.g.*, moose). (b)-(c) The uncertainty measurement with and without virtual outlier training. The in-distribution data $\mathbf{x} \in \mathcal{X} = \mathbb{R}^2$ is sampled from a Gaussian mixture model). Regularizing the model with virtual outliers (c) better captures the OOD uncertainty than without (b).

Alongside, we propose a novel unknown-aware training objective, which contrastively shapes the uncertainty surface between the ID data and synthesized outliers (Section 3.2). During training, VOS simultaneously performs the ID task (*e.g.*, classification or object detection) as well as the OOD uncertainty regularization. During inference time, the uncertainty estimation branch produces a larger probabilistic score for ID data and vice versa, which enables effective OOD detection (Section 3.3).

VOS offers several compelling advantages compared to existing solutions. **(1)** VOS is a *general* learning framework that is effective for both object detection and image classification tasks, whereas previous methods were primarily driven by image classification. Image-level detection can be limiting as an image could be OOD in certain regions while being in-distribution elsewhere. Our work bridges a critical research gap since OOD detection for object detection is timely yet underexplored in literature. **(2)** VOS enables *adaptive* outlier synthesis, which can be flexibly and conveniently used for any ID data without manual data collection or cleaning. In contrast, previous methods using outlier exposure (Hendrycks et al., 2019) require an auxiliary image dataset that is sufficiently diverse, which can be arguably prohibitive to obtain. Moreover, one needs to perform careful data cleaning to ensure the auxiliary outlier dataset does not overlap with ID data. **(3)** VOS synthesizes outliers that can estimate a compact decision boundary between ID and OOD data. In contrast, existing solutions use outliers that are either too trivial to regularize the OOD estimator, or too hard to be separated from ID data, resulting in sub-optimal performance. Our key contributions and results are summarized as follows:

- We propose a new framework VOS addressing a pressing issue—unknown-aware deep learning that optimizes for both ID and OOD performance. VOS establishes *state-of-the-art* results on a challenging object detection task. Compared to the best method, VOS reduces the FPR95 by up to 9.36% while preserving the accuracy on the ID task.

- We conduct extensive ablations and reveal important insights by contrasting different outlier synthesis approaches. We show that VOS is more advantageous than generating outliers directly in the high-dimensional pixel space (*e.g.*, using GAN (Lee et al., 2018a)) or using noise as outliers.

- We comprehensively evaluate our method on common OOD detection benchmarks, along with a more challenging yet underexplored task in the context of object detection. Our effort facilitates future research to evaluate OOD detection in a real-world setting.

## 2 PROBLEM SETUP

We start by formulating the problem of OOD detection in the setting of object detection. Our framework can be easily generalized to image classification when the bounding box is the entire image (see Section 4.2). Most previous formulations of OOD detection treat entire images as anomalies, which can lead to ambiguity shown in Figure 1. In particular, natural images are composed of numerous objects and components. Knowing which regions of an image are anomalous could allow for safer handling of unfamiliar objects. This setting is more realistic in practice, yet also more challenging as it requires reasoning OOD uncertainty at the fine-grained object level.

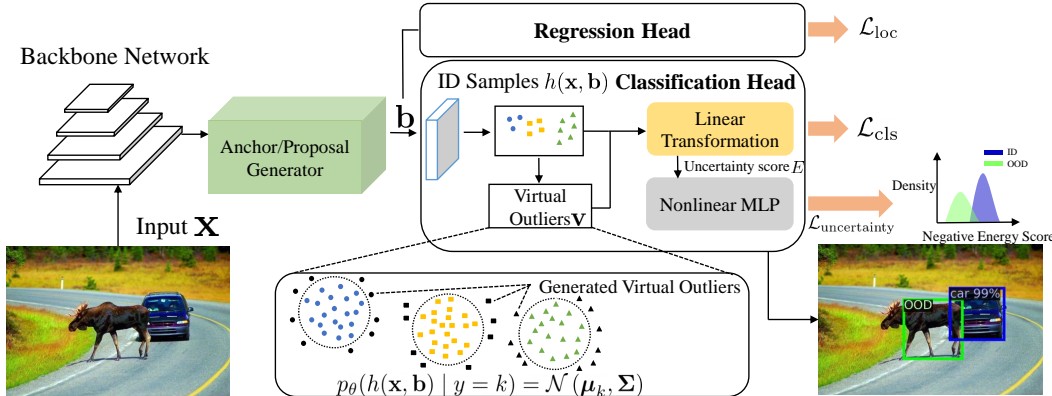

Figure 2: The framework of VOS. We model the feature representation of ID objects as class-conditional Gaussians, and sample virtual outliers $\mathbf{v}$ from the low-likelihood region. The virtual outliers, along with the ID objects, are used to produce the uncertainty loss for regularization. The uncertainty estimation branch ($\mathcal{L}_{\text{uncertainty}}$) is jointly trained with the object detection loss ($\mathcal{L}_{\text{loc}}, \mathcal{L}_{\text{cls}}$).

Specifically, we denote the input and label space by $\mathcal{X} = \mathbb{R}^d$ and $\mathcal{Y} = \{1, 2, ..., K\}$, respectively. Let $\mathbf{x} \in \mathcal{X}$ be the input image, $\mathbf{b} \in \mathbb{R}^4$ be the bounding box coordinates associated with object instances in the image, and $y \in \mathcal{Y}$ be the semantic label for $K$-way classification. An object detection model is trained on in-distribution data $\mathcal{D} = \{(\mathbf{x}_i, \mathbf{b}_i, y_i)\}_{i=1}^{N}$ drawn from an unknown joint distribution $\mathcal{P}$. We use neural networks with parameters $\theta$ to model the bounding box regression $p_\theta(\mathbf{b}|\mathbf{x})$ and the classification $p_\theta(y|\mathbf{x}, \mathbf{b})$.

The OOD detection can be formulated as a binary classification problem, which distinguishes between the in- vs. out-of-distribution objects. Let $P_\mathcal{X}$ denote the marginal probability distribution on $\mathcal{X}$. Given a test input $\mathbf{x}^* \sim P_\mathcal{X}$, as well as an object instance $\mathbf{b}^*$ predicted by the object detector, the goal is to predict $p_\theta(g|\mathbf{x}^*, \mathbf{b}^*)$. We use $g = 1$ to indicate a detected object being in-distribution, and $g = 0$ being out-of-distribution, with semantics outside the support of $\mathcal{Y}$.

## 3 METHOD

Our novel unknown-aware learning framework is illustrated in Figure 2. Our framework encompasses three novel components and addresses the following questions: (1) how to synthesize the virtual outliers (Section 3.1), (2) how to leverage the synthesized outliers for effective model regularization (Section 3.2), and (3) how to perform OOD detection during inference time (Section 3.3)?

### 3.1 VOS: VIRTUAL OUTLIER SYNTHESIS

Our framework VOS generates virtual outliers for model regularization, without relying on external data. While a straightforward idea is to train generative models such as GANs (Goodfellow et al., 2014; Lee et al., 2018a), synthesizing images in the high-dimensional *pixel space* can be difficult to optimize. Instead, our key idea is to synthesize virtual outliers in the *feature space*, which is more tractable given lower dimensionality. Moreover, our method is based on a discriminatively trained classifier in the object detector, which circumvents the difficult optimization process in training generative models.

Specifically, we assume the feature representation of object instances forms a class-conditional multivariate Gaussian distribution (see Figure 3):

$$p_\theta(h(\mathbf{x}, \mathbf{b})|y = k) = \mathcal{N}(\boldsymbol{\mu}_k, \boldsymbol{\Sigma}),$$

where $\boldsymbol{\mu}_k$ is the Gaussian mean of class $k \in \{1, 2, .., K\}$, $\boldsymbol{\Sigma}$ is the tied covariance matrix, and $h(\mathbf{x}, \mathbf{b}) \in \mathbb{R}^m$ is the latent representation of an object instance $(\mathbf{x}, \mathbf{b})$. To extract the latent representation, we use the penultimate layer of the neural network. The dimensionality $m$ is significantly smaller than the input dimension $d$.

To estimate the parameters of the class-conditional Gaussian, we compute empirical class mean $\widehat{\boldsymbol{\mu}}_k$ and covariance $\widehat{\boldsymbol{\Sigma}}$ of training samples $\{(\mathbf{x}_i, \mathbf{b}_i, y_i)\}_{i=1}^N$:

$$\widehat{\boldsymbol{\mu}}_k = \frac{1}{N_k} \sum_{i:y_i=k} h(\mathbf{x}_i, \mathbf{b}_i) \tag{1}$$

$$\widehat{\boldsymbol{\Sigma}} = \frac{1}{N} \sum_k \sum_{i:y_i=k} \left( h(\mathbf{x}_i, \mathbf{b}_i) - \widehat{\boldsymbol{\mu}}_k \right) \left( h(\mathbf{x}_i, \mathbf{b}_i) - \widehat{\boldsymbol{\mu}}_k \right)^\top, \tag{2}$$

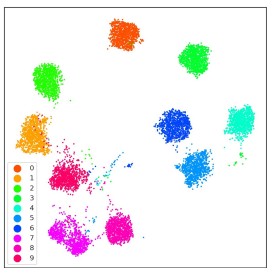

Figure 3: UMAP visualization of feature embeddings of PASCAL-VOC (on a subset of 10 classes).

where $N_k$ is the number of objects in class $k$, and $N$ is the total number of objects. We use online estimation for efficient training, where we maintain a class-conditional queue with $|Q_k|$ object instances from each class. In each iteration, we enqueue the embeddings of objects to their corresponding class-conditional queues, and dequeue the same number of object embeddings.

**Sampling from the feature representation space.** We propose sampling the virtual outliers from the feature representation space, using the multivariate distributions estimated above. Ideally, these virtual outliers should help estimate a more compact decision boundary between ID and OOD data. To achieve this, we propose sampling the virtual outliers $\mathcal{V}_k$ from the $\epsilon$-likelihood region of the estimated class-conditional distribution:

$$\mathcal{V}_k = \{\mathbf{v}_k | \frac{1}{(2\pi)^{m/2}|\widehat{\boldsymbol{\Sigma}}|^{1/2}} \exp\left( -\frac{1}{2}(\mathbf{v}_k - \widehat{\boldsymbol{\mu}}_k)^\top \widehat{\boldsymbol{\Sigma}}^{-1}(\mathbf{v}_k - \widehat{\boldsymbol{\mu}}_k) \right) < \epsilon\}, \tag{3}$$

where $\mathbf{v}_k \sim \mathcal{N}(\widehat{\boldsymbol{\mu}}_k, \widehat{\boldsymbol{\Sigma}})$ denotes the sampled virtual outliers for class $k$, which are in the sublevel set based on the likelihood. $\epsilon$ is sufficiently small so that the sampled outliers are near class boundary.

**Classification outputs for virtual outliers.** For a given sampled virtual outlier $\mathbf{v} \in \mathbb{R}^m$, the output of the classification branch can be derived through a linear transformation:

$$f(\mathbf{v}; \theta) = W_{\text{cls}}^\top \mathbf{v}, \tag{4}$$

where $W_{\text{cls}} \in \mathbb{R}^{m \times K}$ is the weight of the last fully connected layer. We proceed with describing how to regularize the output of virtual outliers for improved OOD detection.

## 3.2 UNKNOWN-AWARE TRAINING OBJECTIVE

We now introduce a new training objective for unknown-aware learning, leveraging the virtual outliers in Section 3.1. The key idea is to perform visual recognition task while regularizing the model to produce a low OOD score for ID data, and a high OOD score for the synthesized outlier.

**Uncertainty regularization for classification.** For simplicity, we first describe the regularization in the multi-class classification setting. The regularization loss should ideally optimize for the separability between the ID vs. OOD data under some function that captures the data density. However, directly estimating $\log p(\mathbf{x})$ can be computationally intractable as it requires sampling from the entire space $\mathcal{X}$. We note that the log partition function $E(\mathbf{x}; \theta) := -\log \sum_{k=1}^K e^{f_k(\mathbf{x};\theta)}$ is proportional to $\log p(\mathbf{x})$ with some unknown factor, which can be seen from the following:

$$p(y|\mathbf{x}) = \frac{p(\mathbf{x}, y)}{p(\mathbf{x})} = \frac{e^{f_y(\mathbf{x};\theta)}}{\sum_{k=1}^K e^{f_k(\mathbf{x};\theta)}},$$

where $f_y(\mathbf{x}; \theta)$ denotes the $y$-th element of logit output corresponding to the label $y$. The negative log partition function is also known as the free energy, which was shown to be an effective uncertainty measurement for OOD detection (Liu et al., 2020a).

Our idea is to explicitly perform a level-set estimation based on the energy function (threshold at 0), where the ID data has negative energy values and the synthesized outlier has positive energy:

$$\mathcal{L}_{\text{uncertainty}} = \mathbb{E}_{\mathbf{v} \sim \mathcal{V}} \; \mathbb{1}\{E(\mathbf{v}; \theta) > 0\} + \mathbb{E}_{\mathbf{x} \sim \mathcal{D}} \; \mathbb{1}\{E(\mathbf{x}; \theta) \leq 0\}$$

This is a simpler objective than estimating density. Since the $0/1$ loss is intractable, we replace it with the binary sigmoid loss, a smooth approximation of the $0/1$ loss, yielding the following:

$$\mathcal{L}_{\text{uncertainty}} = \mathbb{E}_{\mathbf{v}\sim\mathcal{V}}\left[-\log\frac{1}{1+\exp^{-\phi(E(\mathbf{v};\theta))}}\right] + \mathbb{E}_{\mathbf{x}\sim\mathcal{D}}\left[-\log\frac{\exp^{-\phi(E(\mathbf{x};\theta))}}{1+\exp^{-\phi(E(\mathbf{x};\theta))}}\right]. \quad (5)$$

Here $\phi(\cdot)$ is a nonlinear MLP function, which allows learning flexible energy surface. The learning process shapes the uncertainty surface, which predicts high probability for ID data and low probability for virtual outliers $\mathbf{v}$. Liu et al. (2020a) employed energy for model uncertainty regularization, however, the loss function is based on the squared hinge loss and requires tuning two margin hyperparameters. In contrast, our uncertainty regularization loss is completely *hyperparameter-free* and is much easier to use in practice. Moreover, VOS produces probabilistic score for OOD detection, whereas Liu et al. (2020a) relies on non-probabilistic energy score.

**Object-level energy score.** In case of object detection, we can replace the image-level energy with object-level energy score. For ID object $(\mathbf{x},\mathbf{b})$, the energy is defined as:

$$E(\mathbf{x},\mathbf{b};\theta) = -\log\sum_{k=1}^{K} w_k \cdot \exp^{f_k((\mathbf{x},\mathbf{b});\theta)}, \quad (6)$$

where $f_k((\mathbf{x},\mathbf{b});\theta) = W_{\text{cls}}^\top h(\mathbf{x},\mathbf{b})$ is the logit output for class $k$ in the classification branch. The energy score for the virtual outlier can be defined in a similar way as above. In particular, we will show in Section 4 that a learnable $\mathbf{w}$ is more flexible than a constant $\mathbf{w}$, given the inherent class imbalance in object detection datasets. Additional analysis on $w_k$ is in Appendix G.

**Overall training objective.** In the case of object detection, the overall training objective combines the standard object detection loss, along with a regularization loss in terms of uncertainty:

$$\min_{\theta}\mathbb{E}_{(\mathbf{x},\mathbf{b},y)\sim\mathcal{D}}\ [\mathcal{L}_{\text{cls}} + \mathcal{L}_{\text{loc}}] + \beta\cdot\mathcal{L}_{\text{uncertainty}}, \quad (7)$$

where $\beta$ is the weight of the uncertainty regularization. $\mathcal{L}_{\text{cls}}$ and $\mathcal{L}_{\text{loc}}$ are losses for classification and bounding box regression, respectively. This can be simplified to classification task without $\mathcal{L}_{\text{loc}}$. We provide ablation studies in Section 4.1 demonstrating the superiority of our loss function.

## 3.3 Inference-time OOD Detection

During inference, we use the output of the logistic regression uncertainty branch for OOD detection. In particular, given a test input $\mathbf{x}^*$, the object detector produces a bounding box prediction $\mathbf{b}^*$. The OOD uncertainty score for the predicted object $(\mathbf{x}^*,\mathbf{b}^*)$ is given by:

$$p_\theta(g \mid \mathbf{x}^*,\mathbf{b}^*) = \frac{\exp^{-\phi(E(\mathbf{x}^*,\mathbf{b}^*))}}{1+\exp^{-\phi(E(\mathbf{x}^*,\mathbf{b}^*))}}. \quad (8)$$

For OOD detection, one can exercise the thresholding mechanism to distinguish between ID and OOD objects:

$$G(\mathbf{x}^*,\mathbf{b}^*) = \begin{cases} 1 & \text{if } p_\theta(g \mid \mathbf{x}^*,\mathbf{b}^*) \geq \gamma, \\ 0 & \text{if } p_\theta(g \mid \mathbf{x}^*,\mathbf{b}^*) < \gamma. \end{cases} \quad (9)$$

The threshold $\gamma$ is typically chosen so that a high fraction of ID data (e.g., 95%) is correctly classified. Our framework VOS is summarized in Algorithm 1.

---

**Algorithm 1** VOS: Virtual Outlier Synthesis for OOD detection

---

**Input:** ID data $\mathcal{D} = \{(\mathbf{x}_i,\mathbf{b}_i,y_i)\}_{i=1}^{N}$, randomly initialized detector with parameter $\theta$, queue size $|Q_k|$ for Gaussian density estimation, weight for uncertainty regularization $\beta$, and $\epsilon$.
**Output:** Object detector with parameter $\theta^*$, and OOD detector $G$.
**while** *train* **do**
  Update the ID queue $Q_k$ with the training objects $\{(\mathbf{x},\mathbf{b},y)\}$.
  Estimate the multivariate distributions based on ID training objects using Equation 1 and 2.
  Sample virtual outliers $\mathbf{v}$ using Equation 3.
  Calculate the regularization loss using Equation 5, update the parameters $\theta$ based on Equation 7.
**end**
**while** *eval* **do**
  Calculate the OOD uncertainty score using Equation 8.
  Perform thresholding comparison using Equation 9.
**end**

---

| In-distribution $\mathcal{D}$ | Method | FPR95 ↓ | AUROC ↑ | mAP (ID)↑ |
|---|---|---|---|---|
| | | | OOD: MS-COCO / OpenImages | |
| **PASCAL-VOC** | MSP (Hendrycks & Gimpel, 2017) | 70.99 / 73.13 | 83.45 / 81.91 | 48.7 |
| | ODIN (Liang et al., 2018) | 59.82 / 63.14 | 82.20 / 82.59 | 48.7 |
| | Mahalanobis (Lee et al., 2018b) | 96.46 / 96.27 | 59.25 / 57.42 | 48.7 |
| | Energy score (Liu et al., 2020a) | 56.89 / 58.69 | 83.69 / 82.98 | 48.7 |
| | Gram matrices (Sastry & Oore, 2020) | 62.75 / 67.42 | 79.88 / 77.62 | 48.7 |
| | Generalized ODIN (Hsu et al., 2020) | 59.57 / 70.28 | 83.12 / 79.23 | 48.1 |
| | CSI (Tack et al., 2020) | 59.91 / 57.41 | 81.83 / 82.95 | 48.1 |
| | GAN-synthesis (Lee et al., 2018a) | 60.93 / 59.97 | 83.67 / 82.67 | 48.5 |
| | **VOS**-ResNet50 (ours) | **47.53**±2.9 / 51.33±1.6 | 88.70±1.2 / 85.23± 0.6 | 48.9±0.2 |
| | **VOS**-RegX4.0 (ours) | 47.77±1.1 / **48.33**±1.6 | **89.00**±0.4 / **87.59**±0.2 | **51.6**±0.1 |
| **Berkeley DeepDrive-100k** | MSP (Hendrycks & Gimpel, 2017) | 80.94 / 79.04 | 75.87 / 77.38 | 31.2 |
| | ODIN (Liang et al., 2018) | 62.85 / 58.92 | 74.44 / 76.61 | 31.2 |
| | Mahalanobis (Lee et al., 2018b) | 57.66 / 60.16 | 84.92 / 86.88 | 31.2 |
| | Energy score (Liu et al., 2020a) | 60.06 / 54.97 | 77.48 / 79.60 | 31.2 |
| | Gram matrices (Sastry & Oore, 2020) | 60.93 / 77.55 | 74.93 / 59.38 | 31.2 |
| | Generalized ODIN (Hsu et al., 2020) | 57.27 / 50.17 | 85.22 / 87.18 | 31.8 |
| | CSI (Tack et al., 2020) | 47.10 / 37.06 | 84.09 / 87.99 | 30.6 |
| | GAN-synthesis (Lee et al., 2018a) | 57.03 / 50.61 | 78.82 / 81.25 | 31.4 |
| | **VOS**-ResNet50 (ours) | 44.27±2.0 / 35.54±1.7 | 86.87±2.1 / 88.52±1.3 | 31.3±0.0 |
| | **VOS**-RegX4.0 (ours) | **36.61**±0.9 / **27.24**±1.3 | **89.08**±0.6 / **92.13**±0.5 | **32.5**±0.1 |

Table 1: **Main results.** Comparison with competitive out-of-distribution detection methods. All baseline methods are based on a model trained on **ID data only** using ResNet-50 as the backbone, without using any real outlier data. ↑ indicates larger values are better and ↓ indicates smaller values are better. All values are percentages. **Bold** numbers are superior results. We report standard deviations estimated across 3 runs. RegX4.0 denotes the backbone of RegNetX-4.0GF (Radosavovic et al., 2020) for the object detector.

## 4 EXPERIMENTAL RESULTS

In this section, we present empirical evidence to validate the effectiveness of VOS on several real-world tasks, including both object detection (Section 4.1) and image classification (Section 4.2).

### 4.1 EVALUATION ON OBJECT DETECTION

**Experimental details.** We use **PASCAL VOC**[1] (Everingham et al., 2010) and Berkeley DeepDrive (**BDD-100k**[2]) (Yu et al., 2020) datasets as the ID training data. For both tasks, we evaluate on two OOD datasets that contain subset of images from: **MS-COCO** (Lin et al., 2014) and **OpenImages** (validation set) (Kuznetsova et al., 2020). We manually examine the OOD images to ensure they do not contain ID category. We have open-sourced our benchmark data that allows the community to easily evaluate future methods on object-level OOD detection.

We use the Detectron2 library (Girshick et al., 2018) and train on two backbone architectures: ResNet-50 (He et al., 2016) and RegNetX-4.0GF (Radosavovic et al., 2020). We employ a two-layer MLP with a ReLU nonlinearity for $\phi$ in Equation 5, with hidden layer dimension of 512. For each in-distribution class, we use 1,000 samples to estimate the class-conditional Gaussians. Since the threshold $\epsilon$ can be infinitesimally small, we instead choose $\epsilon$ based on the $t$-th smallest likelihood in a pool of 10,000 samples (per-class), generated from the class-conditional Gaussian distribution. A larger $t$ corresponds to a larger threshold $\epsilon$. As shown in Table 6, a smaller $t$ yields good performance. We set $t = 1$ for all our experiments. *Extensive details on the datasets are described in Appendix A, along with a comprehensive sensitivity analysis of each hyperparameter (including the queue size $|Q_k|$, coefficient $\beta$, and threshold $\epsilon$) in Appendix C.*

**Metrics.** For evaluating the OOD detection performance, we report: (1) the false positive rate (FPR95) of OOD samples when the true positive rate of ID samples is at 95%; (2) the area under the receiver operating characteristic curve (AUROC). For evaluating the object detection performance on the ID task, we report the common metric of mAP.

**VOS outperforms existing approaches.** In Table 1, we compare VOS with competitive OOD detection methods in literature. For a fair comparison, all the methods only use ID data without using auxiliary outlier dataset. Our proposed method, VOS, outperforms competitive baselines, including Maximum Softmax Probability (Hendrycks & Gimpel, 2017), ODIN (Liang et al., 2018), energy

---

[1]PASCAL-VOC consists of the following ID labels: Person, Car, Bicycle, Boat, Bus, Motorbike, Train, Airplane, Chair, Bottle, Dining Table, Potted Plant, TV, Sofa, Bird, Cat, Cow, Dog, Horse, Sheep.

[2]BDD-100k consists of ID labels: Pedestrian, Rider, Car, Truck, Bus, Train, Motorcycle, Bicycle, Traffic light, Traffic sign.

| | Method | AUROC ↑ | mAP ↑ |
|---|---|---|---|
| **Image synthesis** | °GAN (Lee et al., 2018a) | 83.67 | 48.5 |
| | °Mixup (Zhang et al., 2018) (mixing ratio 0.4) | 61.23 | 44.3 |
| | °Mixup (Zhang et al., 2018) (mixing ratio 1) | 63.99 | 46.9 |
| **Noise as outliers** | ♮Additive Gaussian noise to ID features | 68.02 | 48.7 |
| | ♮Trainable noise added to the ID features | 66.67 | 48.6 |
| | ♮Gaussian noise | 85.98 | 48.5 |
| **Negative proposals** | ♣All negative proposals | 63.45 | 48.1 |
| | ♣Random negative proposals | 66.03 | 48.5 |
| | ♣Proposals with large background prob (Joseph et al., 2021) | 77.26 | 48.5 |
| | **VOS** (ours) | **88.70** | 48.9 |

Table 2: Ablation on outlier synthesis approaches (on backbone of ResNet-50, COCO is the OOD data).

score (Liu et al., 2020a), Mahalanobis distance (Lee et al., 2018b), Generalized ODIN (Hsu et al., 2020), CSI (Tack et al., 2020) and Gram matrices (Sastry & Oore, 2020). These approaches rely on a classification model trained primarily for the ID classification task, and can be naturally extended to the object detection model due to the existence of a classification head. The comparison precisely highlights the benefits of incorporating synthesized outliers for model regularization.

Closest to our work is the GAN-based approach for synthesizing outliers (Lee et al., 2018a). Compare to GAN-synthesis, VOS improves the OOD detection performance (FPR95) by **12.76**% on BDD-100k and **13.40**% on Pascal VOC (COCO as OOD). Moreover, we show in Table 1 that VOS achieves stronger OOD detection performance while preserving a high accuracy on the original in-distribution task (measured by mAP). This is in contrast with CSI, which displays degradation, with mAP decreased by 0.7% on BDD-100k. Details of reproducing baselines are in Appendix E.

**Ablation on outlier synthesis approaches.** We compare VOS with different synthesis approaches in Table 2. Specifically, we consider three types of synthesis approach: (i°) synthesizing outliers in the pixel space, (ii♮) using noise as outliers, and (iii♣) using negative proposals from RPN as outliers. For type I, we consider GAN-based (Lee et al., 2018a) and mixup (Zhang et al., 2018) methods. The outputs of the classification branch for outliers are forced to be closer to a uniform distribution. For mixup, we consider two different beta distributions $\text{Beta}(0.4)$ and $\text{Beta}(1)$, and interpolate ID objects in the pixel space. For Type II, we use noise perturbation to create virtual outliers. We consider adding fixed Gaussian noise to the ID features, adding trainable noise to the ID features where the noise is trained to push the outliers away from ID features, and using fixed Gaussian noise as outliers. Lastly, for type III, we directly use the negative proposals in the ROI head as the outliers for Equation 5, similar to Joseph et al. (2021). We consider three variants: randomly sampling $n$ negative proposals ($n$ is the number of positive proposals), sampling $n$ negative proposals with a larger probability, and using all the negative proposals. All methods are trained under the same setup, with PASCAL-VOC as in-distribution data and ResNet-50 as the backbone. The loss function is the same as Equation 7 for all variants, with the only difference being the synthesis method.

The results are summarized in Table 2, where VOS outperforms alternative synthesis approaches both in the feature space (♣, ♮) or the pixel space (°). Generating outliers in the pixel space (°) is either unstable (GAN) or harmful for the object detection performance (mixup). Introducing noise (♮), especially using Gaussian noise as outliers is promising. However, Gaussian noise outliers are relatively simple, and may not effectively regularize the decision boundary between ID and OOD as VOS does. Exploiting the negative proposals (♣) is not effective, because they are distributionally close to the ID data.

**Ablation on the uncertainty loss.** We perform ablation on several variants of VOS, trained with different uncertainty loss $\mathcal{L}_{\text{uncertainty}}$. Particularly, we consider: (1) using the squared hinge loss for regularization as in Liu et al., (2) using constant weight $\mathbf{w} = [1, 1, ..., 1]^{\top}$ for energy score in Equation 6, and (3) classifying the virtual outliers as an additional $K + 1$ class in the classification branch. The performance comparison is summarized in Table 3. Compared to the hinge loss, our proposed logistic loss reduces the FPR95 by 10.02% on BDD-100k. While the squared hinge loss in Liu et al. requires tuning the hyperparameters, our uncertainty loss is completely *hyperparameter free*. In addition, we find that a learnable $\mathbf{w}$ for energy score is more desirable than a constant $\mathbf{w}$, given the inherent class imbalance in object detection datasets. Finally, classifying the virtual outliers as

| $\mathcal{D}$ | Method | FPR95 ↓ | AUROC ↑ | object detection mAP (ID) ↑ |
|---|---|---|---|---|
| **PASCAL-VOC** | VOS w/ hinge loss | 49.75 | 87.90 | 46.5 |
| | VOS w/ constant **w** | 51.59 | 88.64 | 48.9 |
| | VOS w/ $K+1$ class | 65.25 | 85.26 | 47.0 |
| | **VOS** (ours) | **47.53** | **88.70** | 48.9 |
| **Berkeley DeepDrive-100k** | VOS w/ hinge loss | 54.29 | 83.47 | 29.5 |
| | VOS w/ constant **w** | 49.25 | 85.35 | 30.9 |
| | VOS w/ $K+1$ class | 52.98 | 85.91 | 30.1 |
| | **VOS** (ours) | **44.27** | **86.87** | 31.3 |

Table 3: **Ablation study.** Comparison with different regularization loss functions (on backbone of ResNet-50, COCO is the OOD data).

an additional class increases the difficulty of object classification, which does not outperform either. This ablation demonstrates the superiority of the uncertainty loss employed by VOS.

**VOS is effective on alternative architecture.** Lastly, we demonstrate that VOS is effective on alternative neural network architectures. In particular, using RegNet (Radosavovic et al., 2020) as backbone yields both better ID accuracy and OOD detection performance. We also explore using intermediate layers for outlier synthesis, where we show using VOS on the penultimate layer is the most effective. This is expected since the feature representations are the most discriminative at deeper layers. We provide details in Appendix F.

**Comparison with training on real outlier data.** We also compare with Outlier Exposure (Hendrycks et al., 2019) (OE). OE serves as a strong baseline since it relies on the *real* outlier data. We train the object detector on PASCAL-VOC using the same architecture ResNet-50, and use the OE objective for the classification branch. The real outliers for OE training are sampled from the OpenImages dataset (Kuznetsova et al., 2020). We perform careful deduplication to ensure there is no overlap between the outlier training data and PASCAL-VOC. Our method achieves OOD detection performance on COCO (AUROC: 88.70%) that favorably matches OE (AUROC: 90.18%), and does not require external data.

## 4.2 EVALUATION ON IMAGE CLASSIFICATION

Going beyond object detection, we show that VOS is also suitable and effective on common image classification benchmark. We use CIFAR-10 (Krizhevsky & Hinton, 2009) as the ID training data, with standard train/val splits. We train on WideResNet-40 (Zagoruyko & Komodakis, 2016) and DenseNet-101 (Huang et al., 2017), where we substitute the object detection loss in Equation 7 with the cross-entropy loss. We evaluate on six OOD datasets: Textures (Cimpoi et al., 2014), SVHN (Netzer et al., 2011), Places365 (Zhou et al., 2018), LSUN-C (Yu et al., 2015), LSUN-Resize (Yu et al., 2015), and iSUN (Xu et al., 2015). The com-

| Method | FPR95 ↓ | AUROC ↑ |
|---|---|---|
| | WideResNet / DenseNet | |
| MSP | 51.05 / 48.73 | 90.90 / 92.46 |
| ODIN | 35.71 / 24.57 | 91.09 / 93.71 |
| Mahalanobis | 37.08 / 36.26 | 93.27 / 87.12 |
| Energy | 33.01 / 27.44 | 91.88 / 94.51 |
| Gram Matrices | 27.33 / 23.13 | 93.00 / 89.83 |
| Generalized ODIN | 39.94 / 26.97 | 92.44 / 93.76 |
| CSI | 35.66 / 47.83 | 92.45 / 85.31 |
| GAN-synthesis | 37.30 / 83.71 | 89.60 / 54.14 |
| **VOS** (ours) | **24.87 / 22.47** | **94.06 / 95.33** |

Table 4: OOD detection results of VOS and comparison with competitive baselines on two architectures: WideResNet-40 and DenseNet-101.

parisons are shown in Table 4, with results averaged over six test datasets. VOS demonstrates competitive OOD detection results on both architectures without sacrificing the ID test classification accuracy (94.84% on pre-trained WideResNet vs. 94.68% using VOS).

## 4.3 QUALITATIVE ANALYSIS

In Figure 4, we visualize the prediction on several OOD images, using object detection models trained without virtual outliers (top) and with VOS (bottom), respectively. The in-distribution data is BDD-100k. VOS performs better in identifying OOD objects (in green) than a vanilla object detector, and reduces false positives among detected objects. Moreover, the confidence score of the false-positive objects of VOS is lower than that of the vanilla model (see the truck in the 3rd column). *Additional visualizations are in Appendix D and H.*

## 5 RELATED WORK

**OOD detection for classification** can be broadly categorized into post hoc and regularization-based approaches. In Bendale & Boult (2016), the OpenMax score is developed for OOD detection based

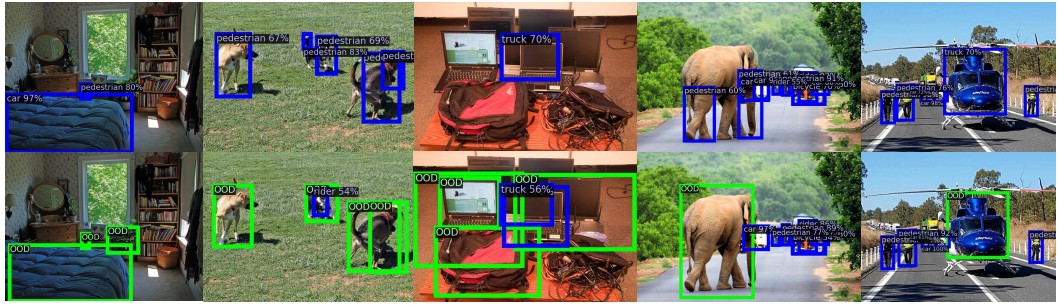

Figure 4: Visualization of detected objects on the OOD images (from MS-COCO) by a vanilla Faster-RCNN (*top*) and VOS (*bottom*). The in-distribution is BDD-100k dataset. **Blue**: Objects detected and classified as one of the ID classes. **Green**: OOD objects detected by VOS, which reduce false positives among detected objects.

on the extreme value theory (EVT). Subsequent work (Hendrycks & Gimpel, 2017) proposed a simple baseline using maximum softmax probability. Improved algorithms have been proposed, such as ensembling (Lakshminarayanan et al., 2017), ODIN (Liang et al., 2018), energy score (Liu et al., 2020a), Mahalanobis distance (Lee et al., 2018b), Gram matrices based score (Sastry & Oore, 2020), and GradNorm score (Huang et al., 2021). Very recently, Sun et al. (2021) showed that a simple activation rectification strategy termed ReAct can significantly improve test-time OOD detection. Theoretical understandings on different post-hoc detection methods are provided in (Morteza & Li, 2022). Different from Lee et al. (2018b), VOS performs dynamic estimation of class-conditional Gaussian during training, which shapes the uncertainty surface over time using our proposed loss.

Another line of approaches explore model regularization using natural outlier images (Hendrycks et al., 2019; Mohseni et al., 2020; Zhang et al., 2021) or images synthesized by GANs (Lee et al., 2018a). However, real outlier data is often infeasible to obtain. Instead, VOS automatically synthesizes virtual outliers which allows greater flexibility and generality. Tack et al. (2020) applied self-supervised learning for OOD detection, which we compare in Section 4. Blum et al. (2021); Jung et al. (2021); Besnier et al. (2021) proposed to detect outliers for semantic segmentation task. Grcic et al. (2021) trained a generative model and synthesize outliers in the pixel space, which cannot be applied to object detection where a scene consists of both known and unknown objects. The regularization is based on entropy maximization, which is different from VOS.

**OOD detection for object detection** is currently underexplored. Joseph et al. (2021) used energy score (Liu et al., 2020a) to identify the OOD data and then labeled them for incremental object detection. In contrast, VOS focuses on OOD detection and adopts a new unknown-aware training objective with a new test-time detection score. Our learning framework is generally applicable to both object detectors and classification models. Moreover, Joseph et al. (2021) used the negative proposals as unknown samples for model regularization, which is suboptimal as we show in Table 2. Harakeh & Waslander (2021); Riedlinger et al. (2021) focused on uncertainty estimation for the localization regression, rather than OOD detection for classification problems. Several works (Dhamija et al., 2020; Miller et al., 2019; 2018; Hall et al., 2020; Deepshikha et al., 2021) used approximate Bayesian methods, such as MC-Dropout (Gal & Ghahramani, 2016) for OOD detection. They require multiple inference passes to generate the uncertainty score, which are computationally expensive on larger datasets and models.

**Open-world object detection** includes out-of-domain generalization (Kim et al., 2021; Wang et al., 2021), zero-shot object detection (Gu et al., 2022; Rahman et al., 2020) and incremental object detection (Liu et al., 2020b; Pérez-Rúa et al., 2020). Most of them either developed measures to mitigate catastrophic forgetting (Joseph et al., 2020) or used auxiliary information (Rahman et al., 2020), such as class attributes to perform object detection on unseen data, which is different from our focus of OOD detection.

# 6 CONCLUSION

In this paper, we propose VOS, a novel unknown-aware training framework for OOD detection. Different from methods that require real outlier data, VOS adaptively synthesizes outliers during training by sampling virtual outliers from the low-likelihood region of the class-conditional distributions. The synthesized outliers meaningfully improve the decision boundary between the ID data and OOD data, resulting in superior OOD detection performance while preserving the performance of the ID task. VOS is effective and suitable for both object detection and classification tasks. We hope our work will inspire future research on unknown-aware deep learning in real-world settings.

## REPRODUCIBILITY STATEMENT

The authors of the paper recognize the importance and value of reproducible research. We summarize our efforts below to facilitate reproducible results:

1. **Datasets.** We use publicly available datasets, which are described in detail in *Section 4.1*, *Section 4.2*, and *Appendix A*.

2. **Baselines.** The description and hyperparameters of the OOD detection baselines are explained in *Appendix E*.

3. **Model training.** Our model training on object detection is based on the publicly available Detectron2 codebase: `https://github.com/facebookresearch/detectron2`. Hyperparamters are specified in *Section 4.1*, with a thorough ablation study provided in *Appendix C*.

4. **Methodology.** Our method is fully documented in Section 3, with the pseudo algorithm detailed in *Algorithm 1*.

5. **Open Source.** The codebase and the dataset will be released for reproducible research. Code is available at `https://github.com/deeplearning-wisc/vos`.

## ETHICS STATEMENT

Our project aims to improve the reliability and safety of modern machine learning models. Our study can lead to direct benefits and societal impacts, particularly for safety-critical applications such as autonomous driving. Our study does not involve any human subjects or violation of legal compliance. We do not anticipate any potentially harmful consequences to our work. Through our study and releasing our code, we hope to raise stronger research and societal awareness towards the problem of out-of-distribution detection in real-world settings.

## ACKNOWLEDGEMENT

Research is supported by Wisconsin Alumni Research Foundation (WARF). We sincerely thank Ziyang (Jack) Cai for helping with inspect the OOD datasets, and members in Li's lab for valuable discussions.

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

# Supplementary Material

## A  EXPERIMENTAL DETAILS

We summarize the OOD detection evaluation task in Table 5. The OOD test dataset is selected from MS-COCO and OpenImages dataset, which contains disjoint labels from the respective ID dataset. The PASCAL model is trained for a total of 18,000 iterations, and the BDD-100k model is trained for 90,000 iterations. We add the uncertainty regularizer (Equation 5) starting from 2/3 of the training. The weight $\beta$ is set to $0.1$. See *detailed ablations on the hyperparameters in Appendix C*.

|  | **Task 1** | **Task 2** |
|---|---|---|
| ID train dataset | VOC train | BDD train |
| ID val dataset | VOC val | BDD val |
| OOD dataset | COCO and OpenImages val | COCO and OpenImages val |
| #ID train images | 16,551 | 69,853 |
| #ID val images | 4,952 | 10,000 |
| #OOD images for COCO | 930 | 1,880 |
| #OOD images for OpenImages | 1,761 | 1,761 |

Table 5: OOD detection evaluation tasks.

## B  SOFTWARE AND HARDWARE

We run all experiments with Python 3.8.5 and PyTorch 1.7.0, using NVIDIA GeForce RTX 2080Ti GPUs.

## C  EFFECT OF HYPERPARAMETERS

Below we perform sensitivity analysis for each important hyperparameter[1]. We use ResNet-50 as the backbone, trained on in-distribution dataset PASCAL-VOC.

**Effect of** $\epsilon$. Since the threshold $\epsilon$ can be infinitesimally small, we instead choose $\epsilon$ based on the $t$-th smallest likelihood in a pool of 10,000 samples (per-class), generated from the class-conditional Gaussian distribution. A larger $t$ corresponds to a larger threshold $\epsilon$. As shown in Table 6, a smaller $t$ yields good performance. We set $t = 1$ for all our experiments.

| $t$ | mAP↑ | FPR95 ↓ | AUROC↑ | AUPR↑ |
|---|---|---|---|---|
| 1 | 48.7 | **54.69** | **83.41** | **92.56** |
| 2 | 48.2 | 57.96 | 82.31 | 88.52 |
| 3 | 48.3 | 62.39 | 82.20 | 88.05 |
| 4 | 48.8 | 69.72 | 80.86 | 89.54 |
| 5 | 48.7 | 57.57 | 78.66 | 88.20 |
| 6 | 48.7 | 74.03 | 78.06 | 91.17 |
| 8 | 48.8 | 60.12 | 79.53 | 92.53 |
| 10 | 47.2 | 76.25 | 74.33 | 90.42 |

Table 6: Ablation study on the number of selected outliers $t$ (per class).

**Effect of queue size** $|Q_k|$. We investigate the effect of ID queue size $|Q_k|$ in Table 7, where we vary $|Q_k| = \{50, 100, 200, 400, 600, 800, 1000\}$. Overall, a larger $|Q_k|$ is more beneficial since the estimation of Gaussian distribution parameters can be more precise. In our experiments, we set the queue size $|Q_k|$ to $1,000$ for PASCAL and 300 for BDD-100k. The queue size is smaller for BDD because some classes have a limited number of object boxes.

**Effect of** $\beta$. As shown in Table 8, a mild value of $\beta$ generally works well. As expected, a large value (e.g., $\beta = 0.5$) will over-regularize the model and harm the performance.

---

[1]Note that our sensitivity analysis uses the speckle noised PASCAL VOC validation dataset as OOD data, which is different from the actual OOD test datasets in use.

| $|Q_k|$ | mAP↑ | FPR95↓ | AUROC↑ | AUPR↑ |
|---|---|---|---|---|
| 50 | 48.6 | 68.42 | 77.04 | 92.30 |
| 100 | 48.9 | 59.77 | 79.96 | 89.18 |
| 200 | 48.8 | 57.80 | 80.20 | 89.92 |
| 400 | 48.9 | 66.85 | 77.68 | 89.83 |
| 600 | 48.5 | 57.32 | 81.99 | 91.07 |
| 800 | 48.7 | **51.43** | 82.26 | 91.80 |
| 1000 | 48.7 | 54.69 | **83.41** | **92.56** |

Table 7: Ablation study on the ID queue size $|Q_k|$.

| $\beta$ | mAP↑ | FPR95↓ | AUROC↑ | AUPR↑ |
|---|---|---|---|---|
| 0.01 | 48.8 | 59.20 | 82.64 | 90.08 |
| 0.05 | 48.9 | 57.21 | 83.27 | 91.00 |
| 0.1 | 48.7 | **54.69** | **83.41** | **92.56** |
| 0.15 | 48.5 | 59.32 | 77.47 | 89.06 |
| 0.5 | 36.4 | 99.33 | 57.46 | 85.25 |

Table 8: Ablation study on regularization weight $\beta$.

**Effect of starting iteration for the regularizer**. Importantly, we show that uncertainty regularization should be added in the middle of the training. If it is added too early, the feature space is not sufficiently discriminative for Gaussian distribution estimation. See Table 9 for the effect of starting iteration $Z$. We use $Z = 12,000$ for the PASCAL-VOC model, which is trained for a total of 18,000 iterations.

| $Z$ | mAP↑ | FPR95↓ | AUROC↑ | AUPR↑ |
|---|---|---|---|---|
| 2000 | 48.5 | 60.01 | 78.55 | 87.62 |
| 4000 | 48.4 | 61.47 | 79.85 | 89.41 |
| 6000 | 48.5 | 59.62 | 79.97 | 89.74 |
| 8000 | 48.7 | 56.85 | 80.64 | 90.71 |
| 10000 | 48.6 | 49.55 | 83.22 | 92.49 |
| 12000 | 48.7 | **54.69** | **83.41** | 92.56 |
| 14000 | 49.0 | 55.39 | 81.37 | **93.00** |
| 16000 | 48.9 | 59.36 | 82.70 | 92.62 |

Table 9: Ablation study on the starting iteration $Z$. Model is trained for a total of 18,000 iterations.

# D  ADDITIONAL VISUALIZATION RESULTS

We provide additional visualization of the detected objects on different OOD datasets with models trained on different in-distribution datasets. The results are shown in Figures 5-8.

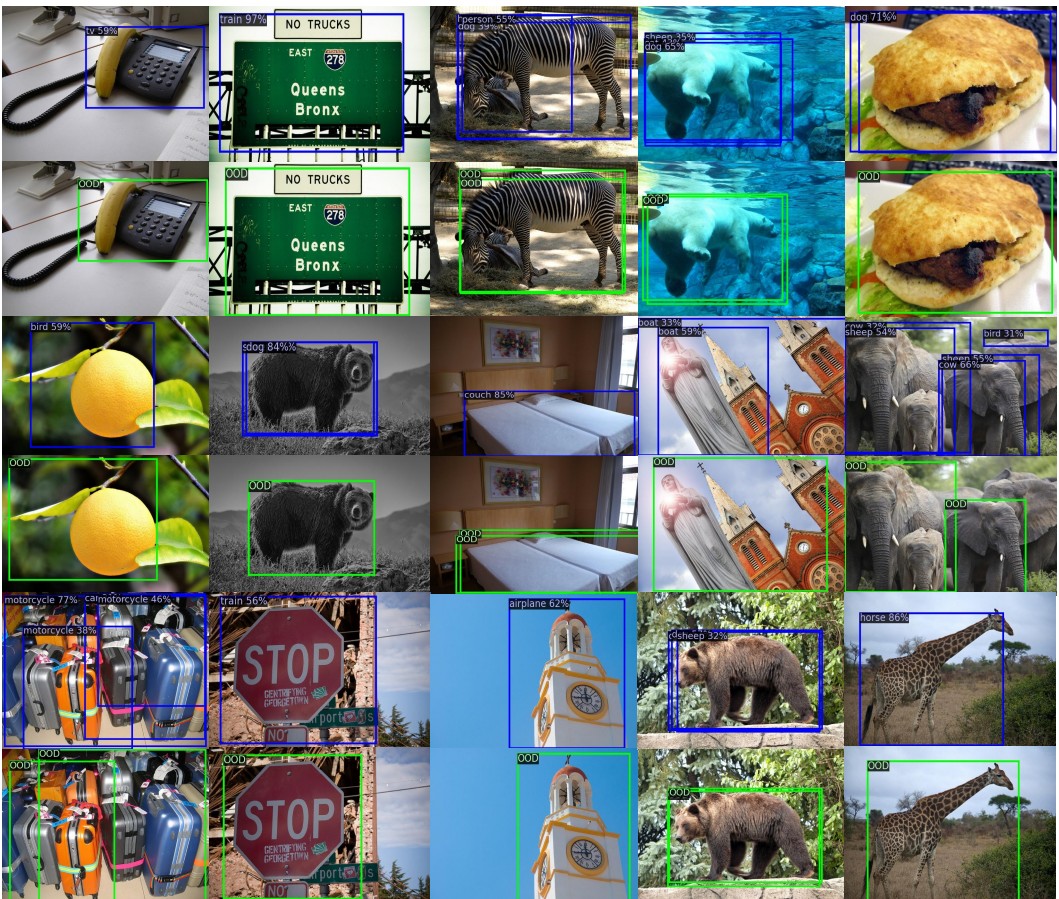

Figure 5: Additional visualization of detected objects on the OOD images (from MS-COCO) by a vanilla Faster-RCNN (*top*) and VOS (*bottom*). The in-distribution is Pascal VOC dataset. **Blue**: Objects detected and classified as one of the ID classes. **Green**: OOD objects detected by VOS, which reduce false positives among detected objects.

# E  BASELINES

To evaluate the baselines, we follow the original methods in MSP (Hendrycks & Gimpel, 2017), ODIN (Liang et al., 2018), Generalized ODIN (Hsu et al., 2020), Mahalanobis distance (Lee et al., 2018b), CSI (Tack et al., 2020), energy score (Liu et al., 2020a) and gram matrices (Sastry & Oore, 2020) and apply them accordingly on the classification branch of the object detectors. For ODIN, the temperature is set to be $T = 1000$ following the original work. For both ODIN and Mahalanobis distance Lee et al. (2018b), the noise magnitude is set to 0 because the region-based object detector is not end-to-end differentiable given the existence of region cropping and ROIAlign. For GAN (Lee et al., 2018a), we follow the original paper and use a GAN to generate OOD images. The prediction of the OOD images/objects is regularized to be close to a uniform distribution, through a KL divergence loss with a weight of 0.1. We set the shape of the generated images to be $100 \times 100$ and resize them to have the same shape as the real images. We optimize the generator and discriminator using Adam (Kingma & Ba, 2015), with a learning rate of 0.001. For CSI (Tack et al., 2020), we use the rotations ($0°, 90°, 180°, 270°$) as the self-supervision task. We set the temperature in the contrastive loss to 0.5. We use the features right before the classification branch (with the dimension to be 1024)

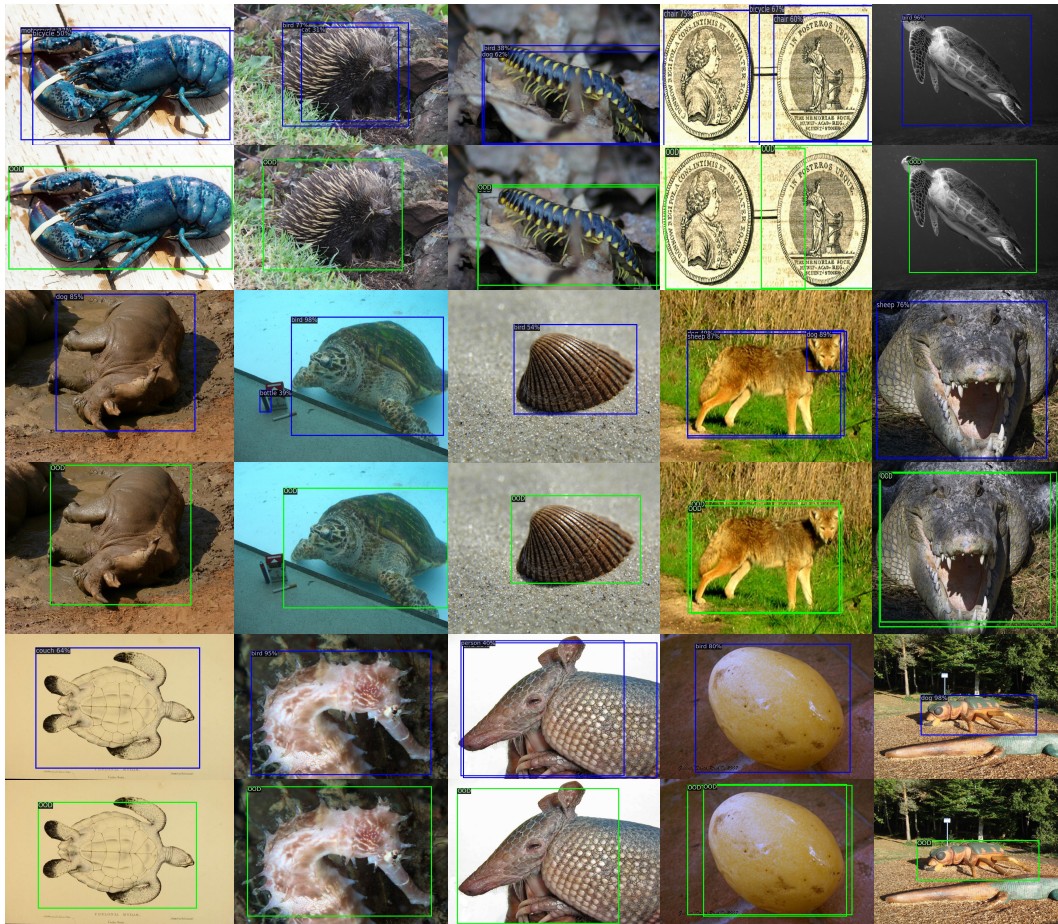

Figure 6: Additional visualization of detected objects on the OOD images (from OpenImages) by a vanilla Faster-RCNN (*top*) and VOS (*bottom*). The in-distribution is Pascal VOC dataset. **Blue**: Objects detected and classified as one of the ID classes. **Green**: OOD objects detected by VOS, which reduce false positives among detected objects.

to perform contrastive learning. The weights of the losses that are used for classifying shifted instances and instance discrimination are both set to 0.1 to prevent training collapse. For Generalized ODIN Hsu et al. (2020), we replace and train the classification head of the object detector by the most effective Deconf-C head shown in the original paper.

## F   VIRTUAL OUTLIER SYNTHESIS USING EARLIER LAYER

In this section, we investigate the effect of using VOS on an earlier layer within the network. Our main results in Table 1 are based on the penultimate layer of the network. Here, we additionally evaluate the performance using the layer before the penultimate layer, with a feature dimension of $1,024$. The results are summarized in Table 10. As observed, synthesizing virtual outliers in the penultimate layer achieves better OOD detection performance than the earlier layer, since the feature representations are more discriminative at deeper layers.

## G   VISUALIZATION OF THE LEARNABLE WEIGHT COEFFICIENT $w$ IN GENERALIZED ENERGY SCORE

To observe whether the learnable weight coefficient $w_k$ in Equation 6 captures dataset-specific statistics during uncertainty regularization, we visualize $w_k$ w.r.t each in-distribution class and the number

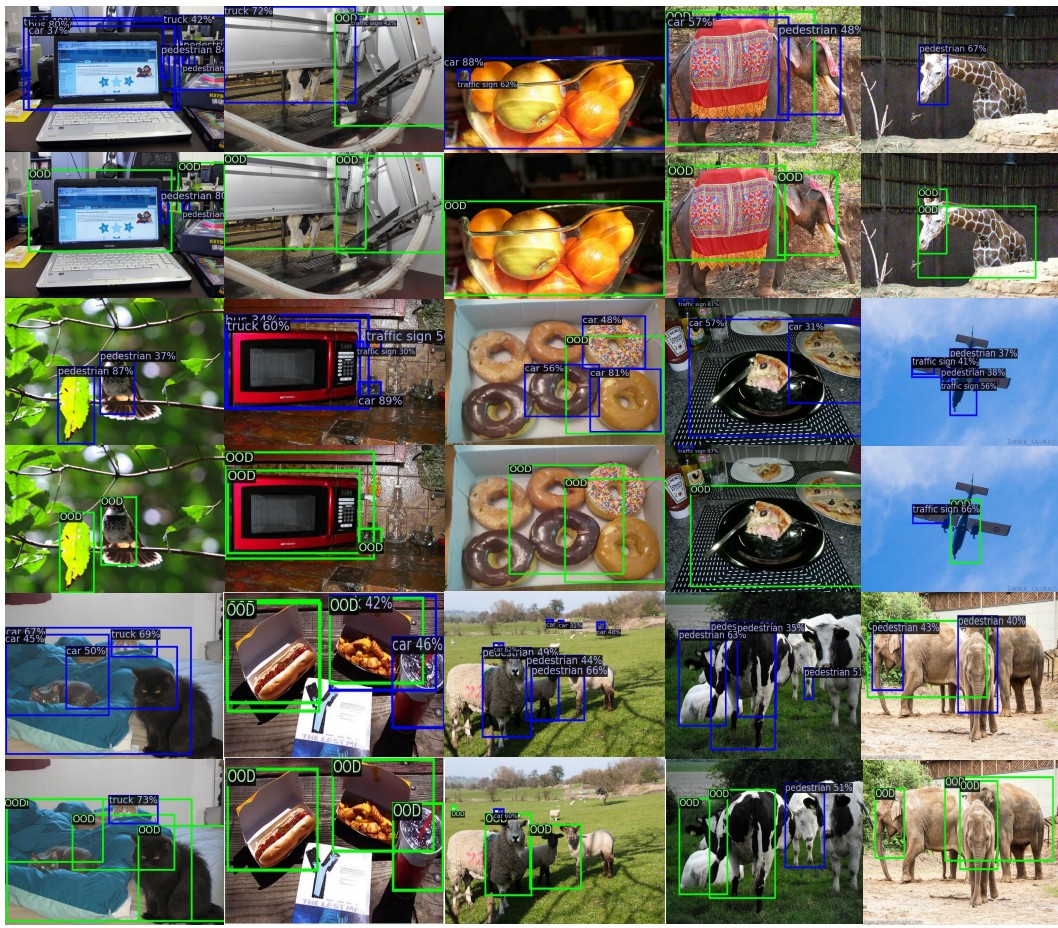

Figure 7: Additional visualization of detected objects on the OOD images (from MS-COCO) by a vanilla Faster-RCNN (*top*) and VOS (*bottom*). The in-distribution is BDD-100k dataset. **Blue**: Objects detected and classified as one of the ID classes. **Green**: OOD objects detected by VOS, which reduce false positives among detected objects.

| Models | FPR95↓ | AUROC↑ | mAP↑ |
|---|---|---|---|
| PASCAL VOC | | | |
| VOS-final | **47.53** | **88.70** | **48.9** |
| VOS-earlier | 50.24 | 88.24 | 48.6 |
| BDD-100k | | | |
| VOS-final | **44.27** | **86.87** | **31.3** |
| VOS-earlier | 49.66 | 86.08 | 30.6 |

Table 10: Performance comparison of employing VOS on different layers. COCO is the OOD data.

of training objects of that class in Figure 9. We use the BDD-100k dataset (Yu et al., 2020) as the in-distribution dataset and the RegNetX-4.0GF (Radosavovic et al., 2020) as the backbone network. As can be observed, the learned weight coefficient displays a consistent trend with the number of training objects per class, which indicates the advantage of using learnable weights rather than constant weight vector with all 1s.

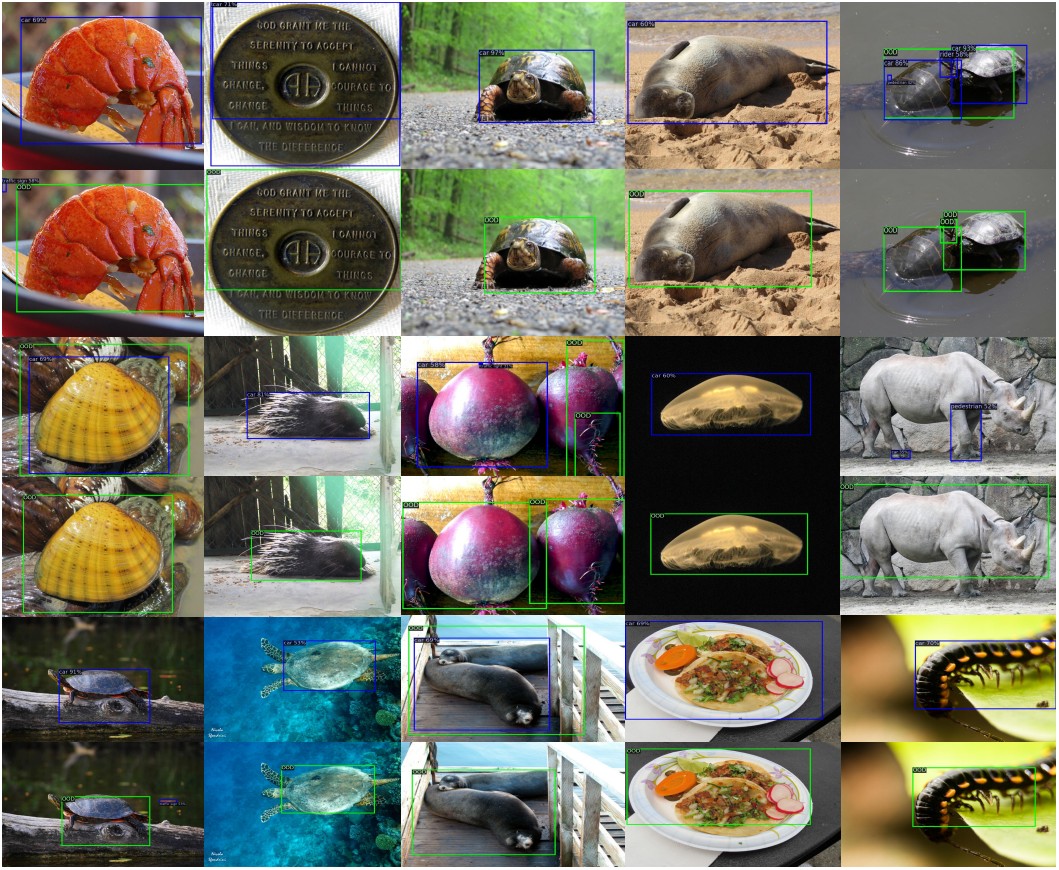

Figure 8: Additional visualization of detected objects on the OOD images (from OpenImages) by a vanilla Faster-RCNN (*top*) and VOS (*bottom*). The in-distribution is BDD-100k dataset. **Blue**: Objects detected and classified as one of the ID classes. **Green**: OOD objects detected by VOS, which reduce false positives among detected objects.

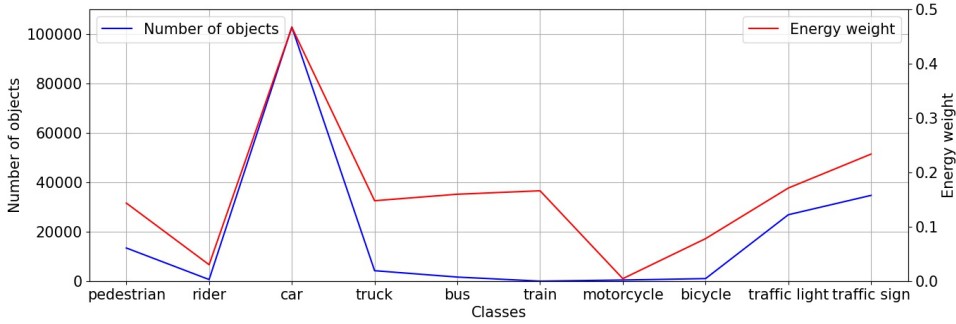

Figure 9: Visualization of learnable weight coefficient in the generalized energy score and the number of training objects per in-distribution class. The value of the weight coefficient is averaged over three different runs.

## H VISUALIZATION OF THE VIRTUAL OUTLIERS

In this section, we visualize the synthesized virtual outliers by VOS using UMAP in Figure 10. The in-distribution dataset is the Pascal VOC dataset with the backbone of ResNet-50. Note that we

cannot visualize virtual outliers in the pixel space since they are synthesized in low-dimensional feature space.

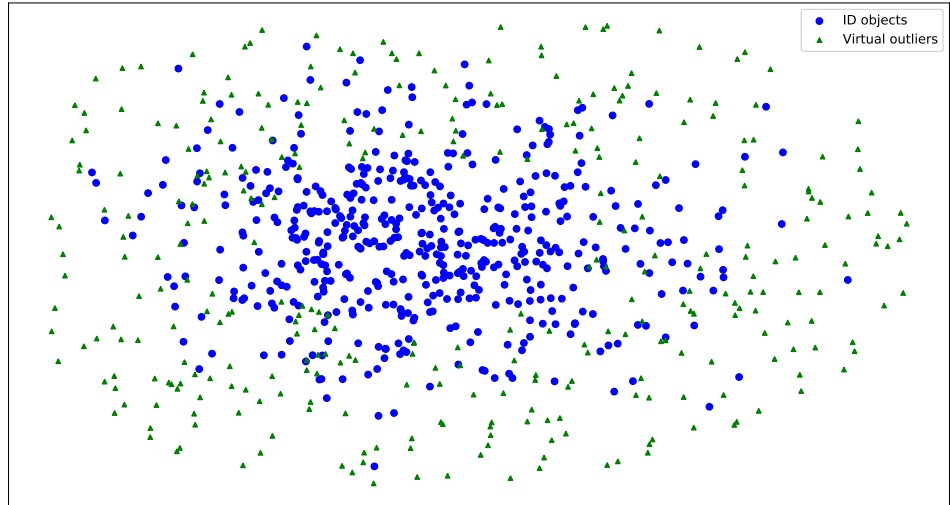

Figure 10: UMAP visualization of the synthesized virtual outliers. The blue points denote the object features from the in-distribution class of `Person`. The green points denote the synthesized virtual outliers from the low-density space *w.r.t* the features from that class.

From Figure 10, the virtual outliers reside in the near-boundary region of the in-distribution feature cluster, which helps the model to learn a compact decision boundary between ID and OOD objects.

## I DISCUSSION ON THE DETECTED, REJECTED AND IGNORED OOD OBJECTS DURING INFERENCE

The focus of VOS is to mitigate the undesirable cases when an OOD object is detected and classified as in-distribution with high confidence. In other words, our goal is to ensure that "if the box is detected, it should be faithfully an in-distribution object rather than OOD". Although generating the bounding box for OOD data is not the focus of this paper, we do notice that VOS can improve the number of boxes detected for OOD data (+25% on BDD trained model compared to the vanilla Faster-RCNN).

The number of OOD objects ignored by RPN can largely depend on the confidence score threshold and the NMS threshold. Hence, we found it more meaningful to compare relatively with the vanilla Faster-RCNN under the same default thresholds. Using BDD100K as the in-distribution dataset and the ResNet as the backbone, VOS can improve the number of detected OOD boxes by 25% (compared to vanilla object detector). VOS also improves the number of rejected OOD samples by 63%.

