# OpenReview forum: "VOS: Learning What You Don't Know by Virtual Outlier Synthesis"
_ICLR.cc/2022/Conference — ICLR 2022 Poster_

### Official Review · Reviewer_XCrg · 2021-10-30

**Correctness:** 4
**Technical Novelty And Significance:** 3
**Empirical Novelty And Significance:** 3
**Recommendation:** 6
**Confidence:** 3

**Main Review:**

Strengths:
1. This work looks solid, and the experiments in this paper are sufficient, which consider almost all key parameters or components of the proposed method and demonstrate the effectiveness of the method. Besides, the method achieves state-of-the-art performance on both object detection and image classification tasks.
2. This paper is well motivated, which aims to utilize synthetic virtual outliers to regularize the model, thus avoiding the collection and cleaning of real OOD data.

Weaknesses:
A few details in the proposed method are not very clear. Specifically, what are the motivation and mechanism of the proposed generalized energy score? How to train the learnable weight coefficients? How to update the ID queue? It would be better to clarify these issues.

Other concerns:
1. Since the real outlier data may not be tightly distributed around the ID data in raw or feature space, why the performance of OE can be viewed as a performance upper bound of VOS?
2. I wonder what virtual outliers look like. It might be better to visualize them.

**Summary Of The Paper:**

This paper proposes an effective method for OOD detection and model regularization, which does not rely on real OOD datasets. Specifically, the proposed method synthesizes virtual outliers by sampling low-likelihood samples in the feature space of class conditional distribution, and adds a novel regularizer to the original ID training objective, which contrastively shapes the uncertainty space between the ID data and synthesized outlier data. Extensive and comprehensive experiments demonstrate the effectiveness of the proposed method.

**Summary Of The Review:**

Overall, this paper is technically and experimentally sound, which conduct comprehensive experiments to verify the proposed method. The reasons to accept outweigh the reasons to reject (e.g., a few unclear details).

---

> ### Author Response · Authors · 2021-11-11
> **Thank you for the constructive feedback**
>
> We are glad that the reviewer found our method well-motivated, solid, and with sufficient experiments. We thank the reviewer for the constructive comments and suggestions, which we address below:
>
> **Discussion on the motivation and mechanism of the generalized energy score**
> Great question raised! The motivation of using learnable weights in the generalized energy score is to compensate for the class imbalance issue. Since the real-world datasets, especially object detection datasets, are usually imbalanced w.r.t different in-distribution classes. A trainable weight in the generalized energy score enables VOS to pay different attention to different in-distribution classes when quantifying the uncertainty. If one in-distribution class contains many more objects in the training dataset, then it should play a larger role when determining whether an object is in-distribution or OOD. **Figure 5** in the appendix verifies the learnable weight can roughly capture the class distribution for the training data, which illustrates the effectiveness of the generalized energy score along with the results in Table 3.
> The learnable weights are randomly initialized and adaptively optimized in the uncertainty regularization loss, where the supervision signal is that the uncertainty score for the in-distribution objects is lower and that for the virtual outliers is higher.
>
> **How to update the ID queue?**
>
> The class-conditional queues are updated following the standard enqueuing and dequeuing operation. In each iteration, we enqueue the embeddings of objects in the current minibatch (to each class-conditional queue) and accordingly dequeue the same number of object embeddings.
>
> We have added more clarifications in the updated version. Thanks for pointing that out!
>
> **Discussion on claiming OE as the upper bound**
> Another great question! OE serves as a strong baseline for us, given the use of *natural image outliers*---which may be closer to the test OOD samples on the natural image manifold. We agree that “upper bound” isn’t an accurate framing and will revise it in the updated version.
>
> **Visualization of the virtual outliers**
> Great suggestion! We have added the visualization in Appendix Section I. The in-distribution dataset is the Pascal VOC dataset with the backbone of ResNet-50. The virtual outliers reside in the near-boundary region of the in-distribution feature clusters, which helps the model to learn a compact decision boundary between ID and OOD objects.

---

### Official Review · Reviewer_rXCn · 2021-10-31

**Correctness:** 3
**Technical Novelty And Significance:** 3
**Empirical Novelty And Significance:** 3
**Recommendation:** 8
**Confidence:** 3

**Main Review:**

Strengths:
(1) This paper is well written and easy to understand. Especially,  the figures deliver the core idea of the whole paper.
(2) The proposed method is straightforward but efficient. Also, the idea could be proved by a toy example.

Weakness:
(1) Some extreme cases are not clearly illustrated. For example, the sampling process is done online. In one batch, it is likely to have only one instance in some class. In that case, I am very concerned about the quality of estimation.
(2) Too many ablations studies are arranged on supplementary files.

Also, I have a question about the unknown-aware training objective. In this part, I understand it like a binary-cross entropy with learnable weight. I am wondering about the influence of learnable weight. I checked the visualization in the supplementary file but do not find the performance effect.



**Summary Of The Paper:**

This paper proposed a novel unknown-aware learning framework dubbed VOS (Virtual Outlier Synthesis), which optimizes the dual objectives of both ID task and OOD detection performance. The key of VOS is to sample outliers in low-dimension feature space instead of generated images. These outliers are used in the training process to help form the compact decision boundary between ID and OOD data.

The extensive experiments show the effectiveness of their method on both the PASCAL-VOC and the DeepDrive-100K datasets.

**Summary Of The Review:**

Overall, this paper could clear deliver the idea to the readers with well-written text and designed figures. The idea is somewhat novelty by the related work. The unknown-aware training object is similar to the dynamic focal loss with the virtual outlier. I agree that the motivation about sampling virtual outliers in the feature space instead of pixel space. Therefore, I vote this paper marginally above the acceptance threshold.

Given that I am not very familiar with OOD, I would like to check other reviewers' opinions.

——————————After Rebutal ——————————

The author has addressed most of my concerns. In such a situation, I would like to raise my scores.

---

> ### Author Response · Authors · 2021-11-11
> **Thank you for the constructive feedback**
>
> We are encouraged that the reviewer finds our work novel, effective, efficient, and with extensive experiments. We are also glad that the reviewer finds our paper well-written and easy to understand. We address comments in detail below:
>
> **Discussion on extreme cases in VOS**
> Thank you for bringing this up for discussion. To clarify, we maintain a class-conditional queue that contains 1,000 samples per class for mean and covariance estimation (Equation 1 and 2). This queue is dynamically updated during training to incorporate new samples and discard old samples (see the third paragraph of Section 3.1). Therefore, this extreme case is not a concern given our queuing mechanism.
>
> We will rearrange and move some ablations to the main paper in the final version with an extra page. Thanks for the suggestion!
>
> **Discussion on the learnable weight in the training objective**
> Great question raised! During the rebuttal phase, we have compared VOS w/o the learnable weight in the logistic regression classifier. The results are summarized in the following table. Pascal VOC is used as the in-distribution dataset and AUROC is reported.
>
> |Method | MS-COCO| OpenImages| mAP|
> |:-----------:|:------------:|:------------:|:------------:|
> | VOS-resnet w/ learnable weight|**88.44** | **83.60**|48.6|
> | VOS-resnet w/o learnable weight| 86.83| 80.83| 47.1|
> | VOS-regnet w/ learnable weight |**89.00** |**83.92**|51.3 |
> | VOS-regnet w/o learnable weight|88.33 | 82.36| 50.4|
>
> The results show that the performance of VOS decreases if the learnable weight in the logistic regression classifier is removed. The learnable weight modulates the slope of the logistic function, which allows learning a sharper binary decision boundary for optimal ID-OOD classification.

---

> > ### Author Response · Authors · 2021-11-27
> > **Thank you for reading our response**
> >
> > Dear Reviewer rXCn,
> >
> > Thank you very much for carefully reading our response and increasing your score! we are glad to hear that our clarification solved your concerns!
> >
> > Best,
> >
> > Authors

---

### Official Review · Reviewer_jkoD · 2021-11-02

**Correctness:** 4
**Technical Novelty And Significance:** 3
**Empirical Novelty And Significance:** 3
**Recommendation:** 8
**Confidence:** 5

**Main Review:**

Strengths:

S1. Sampling artificial training samples in the feature space appears effective.

S2. There is some novelty in the formulations of uncertainty score (learned weights) and outlier probability (sigmoid of modulated uncertainty score).

S3. The authors took time to implement and evaluate several prominent baselines. Experimental evaluation shows competitive performance.

Weaknesses:

W1. Reading the paper requires a lot of guesswork. Figure 2 suggests that the outliers are detected within RPN while in fact the paper considers only ROI classifier (it would be helpful to indicate the location of ROI-align and others layer of the ROI classifier). It is not clear why \Sigma is used instead of \Sigma_k.

W2. The paper does not discuss the computational complexity of recovering \Sigma.

W3. The following work should be cited and discussed since it considers logit normalization for outlier detection:
[R0] Sanghun Jung, Jungsoo Lee, Daehoon Gwak, Sungha Choi, Jaegul Choo. Standardized Max Logits: A Simple yet Effective Approach for Identifying Unexpected Road Obstacles in Urban-Scene Segmentation. ICCV 2021.

W4. The following work should be cited and discussed since it considers learning open-set recognition with artificial outliers:
[R1] Zhi-Lin Zhao, Longbing Cao, Kun-Yu Lin. Revealing Distributional Vulnerability of Explicit Discriminators by Implicit Generators. CoRR abs/2108.09976 (2021)
[R2] Matej Grcic, Petra Bevandic, Sinisa Segvic. Dense Open-set Recognition with Synthetic Outliers Generated by Real NVP. VISIGRAPP (4: VISAPP) 2021: 133-143

Suggestions
- Equation (8) could be clarified by using the notation from (9).
- Would it make sense to include OOD AP in Table 1?
- Propose some intuition on why the present method outperforms [liu20nips] in Table 3. Is it only due to w_k?
- Quantify how many OOD objects are ignored by RPN-a, rejected by (10) and detected as in-distribution objects.

Post rebuttal comment

The revision improves the presentation and clarifies some doubts. Hence I propose to raise my rating to accept.

**Summary Of The Paper:**

The manuscript addresses open-set object detection by modifying the ROI classifier of Faster-RCNN. The probability p(O|x,b) is modeled as sigmoidal activation of a generalized energy score of the classification head logits. The method succeeds if i) RPN ignores the outlier object, or ii) RPN detects the outlier but the candidate gets rejected due to high p(O|x,b).

The generalized energy is formulated as log-sum-weighted-exp-logit in equation (6). The probability p(outlier|x) is modeled as a sigmoid of modulated generalized energy in equations (8) and (9). The classification head is trained to output high energy in inliers and low energy in artificial outliers through equation (8). The artificial outliers are sampled from low-likelihood regions of per-class Gaussians which are fitted to latent features of groundtruth ROIs. These Gaussians are periodically updated during training.

It appears that the authors consider ROI-wide representations of RPN candidates. These representations are likely produced by applying ROI-align and two fully-connected layers to the convolutional features of the RPN. These Gaussian distributions are modelled according to per-class means (\mu_k) i and overall covariance (\Sigma). Virtual outliers are sampled so that their likelihood with respect to \mu_k and \Sigma is less than \epsilon.


**Summary Of The Review:**

The manuscript proposes a reasonable baseline for open-set object detection.

---

> ### Author Response · Authors · 2021-11-11
> **Thank you for the constructive feedback-part I**
>
> We are encouraged that the reviewer found our method competitive and novel. We thank the reviewer for their helpful comments and suggestions, which we address below:
>
> **Clarification on the location of ROI-align and other layers of the ROI classifier**
> We follow the common design of a region-based object detector, i.e, the inputs firstly go through the RPN, and then the ROI classifier (which is the prediction head in Figure 2) outputs the localization and classification predictions.  The virtual outliers are synthesized based on the penultimate layer of the classification branch of the prediction head. **We have updated Figure 2 to make it clearer**.
>
>
> Following [1], we use the same covariance matrix for all the class-conditional Gaussians. This is more efficient than storing K separate covariance matrices. Empirically, we do not observe significant performance differences so we opt for a simplified design choice.
>
> [1] Kimin Lee, Kibok Lee, Honglak Lee, Jinwoo Shin, A Simple Unified Framework for Detecting Out-of-Distribution Samples and Adversarial Attacks, NIPS2018.
>
> **Discussion on the computational complexity of recovering the covariance matrix**
> Excellent suggestion! The covariance matrix is calculated by the multiplication between a matrix of shape [1024, 1000*K] and its transpose, and the computation is negligible using the Pytorch library. K is the number of classes, and 1000 is the number of per-class samples (in the queue we maintain).
>
> **Discussion on more related works**
> Thanks for bringing the literature to our attention! For [R0], it proposes a standardized max-logit approach for detecting outliers in semantic segmentation, which is a **post-hoc** approach. In contrast, VOS employs training-time regularization which better shapes the uncertainty surface with virtual outliers. An illustration is shown in Figure 1, which highlights the uncertainty estimates using the post-hoc approach vs. VOS. [R1] and [R2] both train a generative model and synthesize outliers in the pixel space, which cannot be directly applied to object detection where a scene consists of both known and unknown objects. Their regularization terms are based on entropy maximization, which is different from our energy-based uncertainty regularization objective. We have revised our manuscript and discussed the difference with proper citations in the related work section.
>
> **Experiment results on the OOD AP**
> Great suggestion! There are two average precision metrics in literature: AUPR-In (by treating in-distribution data as positives) vs. AUPR-Out (by treating OOD as positives). From a practical perspective, the score threshold needs to be selected based on the in-distribution data (due to the impossibility to anticipate unknown data ahead of time). For this reason, metrics such as AUPR-Out are less applicable.
>
> We have reported the average precision (AUPR-In) of the baselines and our approach in the following table. The in-distribution data is the BDD100k dataset and the OOD data are COCO and OpenImages dataset. We use a backbone of ResNet-50. The reason for high AP scores is the sample imbalance between the in-distribution and OOD objects. Our evaluation contains significantly more in-distribution objects. For this reason, we believe FPR95 and AUROC are more meaningful metrics to report. Discussions have been included in Appendix H in our updated draft.
>
> |  Method | COCO| OpenImages|
> |:-----------:|:------------:|:------------:|
> | MSP| 99.53 | 99.21|
> |ODIN| 99.38| 99.52|
> |Mahalanobis| 96.66 |  97.26 |
> |Gram matrices | 94.22|95.95 |
> |Energy score|99.53 |99.79 |
> |Generalized ODIN|99.70 | 99.78|
> |CSI|99.69 | 99.65|
> |GAN-synthesis|99.49 |99.20 |
> | VOS| **99.71**| **99.81**|
>
> **Clarification on VOS outperforms training with the energy score**
> The difference between VOS w/ the generalized energy score and VOS w/ constant **w** is whether the weight for each in-distribution class is trainable. Since the real-world datasets, especially object detection datasets, are usually imbalanced w.r.t different in-distribution classes. A trainable weight in the generalized energy score enables VOS to pay different attention to different in-distribution classes when quantifying the uncertainty. If one in-distribution class contains much more objects in the training dataset, then it should play a larger role when determining whether an object is in-distribution or OOD. Figure 5 in the appendix verifies the learnable weight can roughly capture the class distribution for the training data, which illustrates the effectiveness of the generalized energy score along with the results in Table 3.

---

> > ### Author Response · Authors · 2021-11-11
> > **Thank you for the constructive feedback-part II**
> >
> > **Quantify how many OOD objects are ignored by RPN, rejected by VOS, and detected as in-distribution objects**
> > Great question raised! The number of OOD objects ignored by RPN can largely depend on the confidence score threshold and the NMS threshold. Hence, we found it more meaningful to compare relatively with vanilla object detector under the same thresholds (default setting in Faster-RCNN). Using BDD100K as the in-distribution dataset and the ResNet as the backbone, VOS can improve the number of detected OOD boxes by 25\% (compared to vanilla object detector). VOS also improves the # of rejected OOD samples by 63\%. We have updated the draft (see appendix section J). Thanks for pointing that out!
> >
> > **Clarification on the Equation (8)**
> > Equation (8) is mainly used to denote the binary classification loss with the energy score for each object as the input. We have revised our manuscript to clarify the meaning of this equation.

---

> > > ### Comment · Reviewer_jkoD · 2021-11-26
> > > **Additional references, title**
> > >
> > > Thank you on interesting discussion!
> > >
> > > I am leaning towards increasing my score to accept.
> > >
> > > In the meantime I thought of two additional relevant previous works:
> > > [1] Hongjie Zhang, Ang Li, Jie Guo, Yanwen Guo. Hybrid Models for Open Set Recognition. ECCV 2020.
> > > [2] Hermann Blum, Paul-Edouard Sarlin, Juan I. Nieto, Roland Siegwart, Cesar Cadena. The Fishyscapes Benchmark: Measuring Blind Spots in Semantic Segmentation. Int. J. Comput. Vis. 129(11): 3119-3135 (2021)
> > >
> > > Both of these two papers fit generative models to latent representations in order to detect OOD input, although none of them addresses object detection. A brief discussion of these papers in this context may be a worthwhile addition to the present work.
> > >
> > > I find the title a bit too general since the paper addresses only object detection. Perhaps it would be better to mention object detection, eg: UNKNOWN-AWARE OBJECT DETECTION WITH VIRTUAL OUTLIER SYNTHESIS.

---

> > > > ### Author Response · Authors · 2021-11-26
> > > > **Thank you for reading our response**
> > > >
> > > > Dear Reviewer jkoD,
> > > >
> > > > Thank you very much for carefully reading our response and increasing your score!
> > > >
> > > > We agree with you that the mentioned literature is indeed relevant to our work, where the first one proposes a flow-based density estimator for open-set classification and the second one adopts pixel-wise uncertainty estimates for semantic segmentation networks. We will merge those works in the final version with proper citations. Thanks for pointing that out!
> > > >
> > > > For the title, we are aware that VOS can also be applied for classification models (section 4.2) so we intend to use a general term "unknown-aware learning". We will consider clarifying and highlighting the object detection networks in the introduction after revision.
> > > >
> > > > Best,
> > > >
> > > > Authors

---

### Official Review · Reviewer_yx7X · 2021-11-03

**Correctness:** 4
**Technical Novelty And Significance:** 2
**Empirical Novelty And Significance:** 2
**Recommendation:** 5
**Confidence:** 5

**Main Review:**

Strengths:

1. Considering synthesized OOD sample to expose model is more feasible compared to procuring real OOD samples. Authors synthesis the OOD samples in the low-dimensional feature space which is more efficient to generate than the actual image space.

2. Unknown-aware training is also an effective idea to differentiate the in distribution samples from OOD samples.

3.  Experiments are conducted on the benchmark dataset and results are impressive. Extensive experiments and ablation studies are conducted to evaluate various components of the proposed approach.

Weaknesses:

1. Overall, the novelty of the proposed approach is limited. Considering synthesized outliers to detect OOD or novel objects are explored in various previous approaches. Authors consider a multimodal Gaussian distribution to approximate in distribution samples and sample new OOD samples. However, the class boundaries in the high-dimensional space may not be Gaussian.

2. Authors consider the object detector network for generating the bounding boxes for the OOD datasets. In this case, the object detector may miss a significant number of bounding boxes as some of the objects instances may not appear in the in-distribution data. In this case, these boxes will not appear for the OOD test. Do authors see this behavior and have any comments on this issue.

3. In table 2, while comparing the other generative methods, do authors consider the same method for detecting OOD. For example, if the additional loss function is included in the approach, then the comparison may not reflect just the benefit of the proposed synthesis method.

4. In table 1, authors are also encouraged to provide the backbone networks for the methods they compare against so that contribution can be better evaluated.

5. What is the motivation for considering the synthetic OOD samples when real outlier data are abundant, easily available, and achieve better performance.

**Summary Of The Paper:**

This paper presents an approach for detecting out of distribution (OOD) samples by synthesizing outlier samples and considering an unknown-aware learning mechanism. The synthetic outlier samples are used to learn a tighter class boundary for in-distribution samples. The model is trained with an uncertainty-aware loss function that encourages a high probability score in distribution samples and a low probability score for OOD samples.


**Summary Of The Review:**

The authors proposed an OOD detection approach the relies on synthetic outlier samples. The proposed approach is shown to be effective but the overall novelty is limited.

---

> ### Author Response · Authors · 2021-11-11
> **Thank you for the constructive feedback-part I**
>
> We are glad that the reviewer found our method effective with extensive experiments. We thank the reviewer for the constructive comments and suggestions, which we address below:
>
> **Novelty of the approach**
> VOS consists of three novel components tackling challenges of outlier synthesis and effective model regularization with synthesized outliers. The three components work in a synergistic fashion, constituting a new unknow-aware learning framework. We recap our novelty points individually:
>
>   - (1) **Outlier synthesis**: To the best of our knowledge, our work is the first to explore feature-space outlier synthesis for OOD detection. While much research progress is made in OOD detection for classification models, the problem remains underexplored in the context of object detection. We are not aware of any prior literature on synthesizing outliers for the challenging task of object detection. [1] explored using GAN-based synthesis for image-level rather than object-level OOD detection. Synthesizing outliers in the pixel space is not very feasible for object detection, __since an image could contain anomalies in certain specific regions while being in distribution elsewhere__. A key novelty and contribution of VOS is to seek feature-space outlier synthesis, which makes the sampling more tractable. Moreover, our method is applicable broadly on object detection and image classification, supported by thorough ablations that provide insights and understandings on our method (see **Tables 2 and 3**).
>
> - (2) **Unknown-aware training objective**: As recognized by several reviewers, another novelty is the uncertainty-aware learning objective powered by synthesized outliers. We demonstrate the effectiveness of the uncertainty-regularization with synthesized outliers and show superior performance compared to prior state-of-the-art (see Table 1).
>
> - (3) **Test-time OOD detection score**: We also introduce a new test-time OOD detection score (Section 3.3), which produces probabilistic OOD uncertainty estimation. This advances prior work that relies on a non-probabilistic energy score.
>
> Lastly, we believe that “the feature representation of object instances forms a class-conditional multivariate Gaussian distribution” is a reasonable assumption. This has been experimentally verified in Figure 3. A similar assumption is made in [2], which is widely recognized in literature and also assumes the class-conditional Gaussian distributions. Given density estimation in high-dimensional space is a known difficult problem, we view our approach as a reasonable approximation that leads to empirical efficacy.
>
> [1] Kimin Lee, Honglak Lee, Kibok Lee, Jinwoo Shin, Training Confidence-calibrated Classifiers for Detecting Out-of-Distribution Samples, ICLR 2018.
>
> [2] Kimin Lee, Kibok Lee, Honglak Lee, Jinwoo Shin, A Simple Unified Framework for Detecting Out-of-Distribution Samples and Adversarial Attacks, NIPS2018.
>
>
> **Discussion on the missing objects on OOD data**
> Great question raised! Our focus is to mitigate the undesirable cases when an OOD object is detected and classified as in-distribution with high confidence. In other words, our goal is to ensure that “if the box is detected, it should be faithfully an in-distribution object rather than OOD”. Although generating the bounding box for OOD data is not our focus, we do notice that compared to the vanilla faster-rcnn, VOS can improve the number of boxes detected for OOD data (+25% on BDD trained model).
>
> **Clarification on comparison with other synthesis approaches**
> Another excellent question! Yes indeed, we use the same training objective (i.e., the energy-based uncertainty regularization loss) and the same OOD detection score for all the compared variants except for the GAN-based method.
>
> For GAN-synthesis, we follow the loss function in the original paper, which forces the classification outputs of the synthesized images to be close to a uniform distribution. During rebuttal, we have tried to use energy-based uncertainty regularization loss in replacement, the AUROC on the PASCAL VOC dataset (MS-COCO as OOD) is 83.07\%, which is quite similar to the originally reported result 82.91\% and does not outperform VOS (88.44\%). We have revised the draft with the new result in the main paper.
>
> **Clarification on backbone networks for the baselines**
> We use ResNet-50 for all the baseline methods in Table 1 to ensure a fair comparison. This is specified in the **caption of Table 1**.

---

> > ### Author Response · Authors · 2021-11-11
> > **Thank you for the constructive feedback-Part II**
> >
> > **Motivation of considering the synthetic OOD samples**
> > Our motivation is described in the **last paragraph of the introduction**. Using real outlier data is not always feasible or flexible. Practically, we need to perform careful data cleaning to ensure the auxiliary outlier dataset does not overlap with in-distribution data. Moreover, we may need to reconstruct the outlier dataset if the in-distribution data changes or expands to contain overlapping concepts w.r.t the auxiliary data. _These approaches become even more restrictive in the context of object detection, where an image could contain anomalies in certain specific regions while being in distribution elsewhere_. It is arguably expensive to annotate a large number of OOD objects in complex scenes---in addition to the already costly process of ID data (bounding box) collection. In contrast, VOS can adaptively synthesize the outliers and is applicable for object detection.

---

### Author Response · Authors · 2021-11-11
**General response -- thanks to all reviewers for the valuable feedback**

We are pleased to see that reviewers find that the method is **novel**, **solid** and **effective** (R1, R2, R3, R4), **proposes a reasonable baseline for open-set object detection** (R2), and the results on both object detection and image classification tasks are **comprehensive**, **competitive** and **impressive** (R1, R2, R3, R4). We are equally glad that the reviewer found the paper **well-written** and **easy to understand** (R3).

We have addressed the reviewers’ comments and concerns in **individual responses to each reviewer**. The reviews allowed us to improve our draft and the changes made in the revised draft are summarized below:

From Reviewer yx7X:
+ Clarified on novelty
+ Clarified backbone networks, comparison settings, and the motivation of our work.
+ Expanded discussion on OOD box detection (see revision).

From Reviewer jkoD:
+ Revised architecture figure 2.
+ Clarified covariance matrix, and comparison with energy score.
+ Expanded discussion on related works (see revision).
+ Added results on OOD AP, ignored and rejected objects during inference (see Appendix H and J).

From Reviewer rXCn:
+ Expanded discussion on the extreme cases and learnable weights.

From Reviewer XCrg:
+ Clarified the details of the generalized energy score and comparison w.r.t outlier exposure.
+ Clarified queuing update (see revision in Section 3, page 4).
+ Added visualization of the virtual outliers (see Appendix I).

---

### Public Comment · ~Umar_Khalid1 · 2022-02-15
**The Method is not scalable.**

I rerun a few experiments for Image Classification on CIFAR-100 and noted that the performance degrades as the number of classes increases and probably one of the reasons the authors didn't report results on CIFAR-100 and Imagenet. I argue that the results are not SOTA even on CIFAR-10.
Probably, the only contribution is that the paper addresses the OOD in object detection.

---

### Decision · Program_Chairs · 2022-01-20

**Decision:**

Accept (Poster)

**Comment:**

This paper proposes to synthetize virtual outliers by sampling from low-likelihood regions of the feature space of a class conditional distribution, in order to make more robust predictions via a regularization loss term.

In the reviewing phase certain criticisms were raised by reviewers: namely that i) the paper was not clear w.r.t. its goal, motivation and position in the literature of OOD detection for bounding boxes; ii) details about the energy-based formulation and covariance definitions and iii) experimental setting and metric used were missing. During rebuttal the authors answered to all the above criticisms up to a satisfying extent and were able to increase two reviewers' scores.

The paper is accepted conditioned on the fact that the camera-ready includes the additional details and discussions that arose in the comments with a specific emphasis on properly framing (and limiting) the motivation of OOD detection for open-set object detection and it is expected to properly cite the literature of the more general OOD detection task as discussed in the comments.